# Entanglement Hamiltonian and effective temperature of non-Hermitian quantum spin ladders

Pei-Yun Yang[1] and Yu-Chin Tzeng[2⋆]

**1** Department of Physics, National Taiwan University, Taipei 106319, Taiwan
**2** Department of Electrophysics and Center for Theoretical and Computational Physics, National Yang Ming Chiao Tung University, Hsinchu 300093, Taiwan

⋆ yctzeng@nycu.edu.tw

## Abstract

Quantum entanglement plays a crucial role not only in understanding Hermitian many-body systems but also in offering valuable insights into non-Hermitian quantum systems. In this paper, we analytically investigate the entanglement Hamiltonian and entanglement energy spectrum of a non-Hermitian spin ladder using perturbation theory in the biorthogonal basis. Specifically, we examine the entanglement properties between coupled non-Hermitian quantum spin chains. In the strong coupling limit ($J_{\text{rung}} \gg 1$), first-order perturbation theory reveals that the entanglement Hamiltonian closely resembles the single-chain Hamiltonian with renormalized coupling strengths, allowing for the definition of an *ad hoc* temperature. Our findings provide new insights into quantum entanglement in non-Hermitian systems and offer a foundation for developing novel algorithms, such as applying finite-temperature Density Matrix Renormalization Group (DMRG) to non-Hermitian quantum systems.

## 1 Introduction

Quantum entanglement is a foundational concept that significantly enhances our understanding of many-body physics by elucidating quantum correlations between subsystems. The entanglement reveals concealed connections beyond classical physics. [1, 2] Suppose the total Hamiltonian $H = H_A + H_B + H_{AB}$ is written as summation of the subsystem Hamiltonians $H_A$ and $H_B$, and their interaction $H_{AB}$. To further analyze the ground-state entanglement properties, the reduced density matrix $\rho_A = \text{Tr}_B |\psi_0\rangle\langle\psi_0|$ usually becomes an essential tool, where $|\psi_0\rangle$ is the normalized ground-state of the total Hamiltonian $H$, and the partial trace is performed on tracing the degrees of freedom of subsystem B. One common measure for quantifying entanglement between the subsystem A and B is the von Neumann entanglement entropy, defined by $S_{\text{von}} = -\sum_i \omega_i \ln \omega_i$, where $\omega_i$ is the $i$th eigenvalue of $\rho_A$. For example, in gapless systems where the low-energy theory is described by conformal field theory, the ground-state entanglement entropy exhibits a logarithmic scaling behavior with respect to subsystem size. This scaling allows for extraction of the central charge, a key parameter in the conformal field theory, which serves as an indicator of the phase transition's universality class. [3, 4]. For gapped systems, the ground-state entanglement entropy follows an area law, meaning it is proportional to the size of subsystem's boundary.

The entanglement energy spectrum provides a detailed view of the quantum correlations between subsystems, with the entanglement entropy serving as a condensed summary of the information contained within this spectrum. The entanglement energy $\xi_i$ is the $i$th energy eigenvalue of a hypothetical Hamiltonian, called entanglement Hamiltonian $H_E$, which is defined by regarding the reduced density matrix as a thermal density matrix of the entanglement Hamiltonian at unity temperature, $\rho_A = e^{-H_E}/Z$. Where $Z$ is the partition function that ensures $\text{Tr}[\rho_A] = 1$. Although the exact form of $H_E$ is generally unknown, its eigenvalues can be obtained by taking logarithm on the eigenvalues of the reduced density matrix, $\xi_i = -\ln \omega_i - \ln Z$. The entanglement energy spectrum provides deep insights into topological systems and can be considered as a kind of 'fingerprint' of these systems. [5] This 'fingerprint' means that even when the entire system is divided into two halves, a topological system generates a gapless edge state, and this signature can be observed in the low-energy portion of the entanglement energy spectrum. [5–8] For example in the 2-dimensional topological systems, such as the fractional quantum Hall systems, the low-energy portion of momentum-resolved entanglement spectrum presents the same state counting with the low-energy spectrum of the edge Hamiltonian. This relationship is known as the renowned Li-Haldane conjecture [9] or the edge-entanglement spectrum correspondence. [8]

Interesting phenomena related to the entanglement Hamiltonian $H_E$ can also arise when the subsystem B is considered as an ancilla system copied from the subsystem A, and introducing strong enough interaction $H_{AB}$ to create a nearly maximally entangled state between A and B. [10–12] For example, in an antiferromagnetic spin ladder where the rung coupling is much stronger than the leg coupling, the ground-state forms multiple rung-singlets, resulting in a nearly maximally entangled state between the two legs. [13–21] In contrast to the Li-Haldane conjecture in topological systems, the entire entanglement spectrum has some similarity to the energy spectrum of subsystem A. Remarkably, under carefully selected parameters, the entanglement Hamiltonian $H_E \approx \beta H_A$ can be proportional to the Hamiltonian of subsystem A by a constant $\beta$. In other words, the finite temperature properties of an isolated system A can be approximated by the reduced density matrix $\rho_A$ obtained from the ground-state of an enlarged system at zero temperature,

$$\rho_A \approx \frac{1}{Z} \exp[-\beta H_A], \tag{1}$$

where $\beta$ is the inverse temperature as a function of system parameters. This development allows the finite-temperature Density Matrix Renormalization Group (DMRG) based on matrix product states to evolve into a robust numerical method for finite-temperature strongly correlated systems. [10]

In this paper, we show that Eq. (1) remains valid for non-Hermitian Hamiltonians. This implies that the ancilla trick used in finite-temperature algorithms [10–12] is also expected to be effective for non-Hermitian quantum systems. In the next section, we briefly introduce non-Hermitian quantum mechanics and describe the non-Hermitian Hamiltonian for the spin-1/2 ladder.

## 2   non-Hermitian Hamiltonian

Non-Hermitian systems have become an important multidisciplinary field of study, [22–24] spanning photonics, [25] condensed matter physics, [26, 27] and quantum information science. [28–36] Unlike traditional quantum systems, which are governed by Hermitian Hamiltonians that ensure real eigenvalues and physically observable energy levels, non-Hermitian systems are described by Hamiltonians that do not necessarily satisfy this condition. This leads

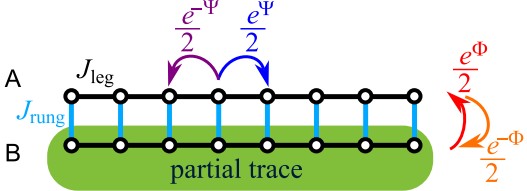

Figure 1: Schematic representation of the non-Hermitian spin-1/2 ladder with nonreciprocal couplings. All parameters are real and positive. The shaded region indicates the part of the system on which the partial trace is performed.

to complex eigenvalues and unique physical phenomena, e.g. exceptional points [37–42] and non-Hermitian skin effect. [43–45] An exceptional point (EP) in non-Hermitian systems occurs when two or more eigenvalues and their corresponding eigenvectors coalesce, rendering the Hamiltonian non-diagonalizable. This leads to unique phenomena, such as enhanced sensitivity to external changes [25] and a negative divergence in real part of fidelity susceptibility [39–41] or quantum metric, [46–48] which measures how ground state changes under perturbations. The non-Hermitian skin effect arises in systems with nonreciprocal coupling. [43] This nonreciprocity means that particles or excitations have different probabilities of hopping forward versus backward. As a result, even in the absence of an external field or disorder, the system can exhibit an accumulation of states at one end under open boundary condition, leading to the Hermitian bulk – non-Hermitian boundary correspondence. [49]

Mathematically, nonreciprocal coupling is often introduced into a system's Hamiltonian through asymmetric hopping terms. [26, 27] For instance, in a one-dimensional lattice model, the hopping amplitude from site $j$ to site $j+1$ might differ from the amplitude from site $j+1$ to site $j$. This asymmetry results in a complex band structure with eigenvalues that can form loops in the complex plane, leading to the skin effect. [49] In this paper, we study the following non-Hermitian Hamiltonian $H = H_A + H_B + H_{AB}$ for spin-1/2 ladder with the nonreciprocal coupling.

$$H_A = J_{\text{leg}} \sum_{j=1}^{N} \left[ \frac{1}{2} \left( e^{\Psi} S_{j,A}^{+} S_{j+1,A}^{-} + e^{-\Psi} S_{j,A}^{-} S_{j+1,A}^{+} \right) + \Delta S_{j,A}^{z} S_{j+1,A}^{z} \right], \tag{2a}$$

$$H_B = J_{\text{leg}} \sum_{j=1}^{N} \left[ \frac{1}{2} \left( e^{\Psi} S_{j,B}^{+} S_{j+1,B}^{-} + e^{-\Psi} S_{j,B}^{-} S_{j+1,B}^{+} \right) + \Delta S_{j,B}^{z} S_{j+1,B}^{z} \right], \tag{2b}$$

$$H_{AB} = J_{\text{rung}} \sum_{j=1}^{N} \left[ \frac{1}{2} \left( e^{\Phi} S_{j,A}^{+} S_{j,B}^{-} + e^{-\Phi} S_{j,A}^{-} S_{j,B}^{+} \right) + \Delta S_{j,A}^{z} S_{j,B}^{z} \right], \tag{2c}$$

where $N$ denotes the number of rungs, and $\Phi$ and $\Psi$ are real parameters that control the nonreciprocal coupling between the legs and within each leg, respectively. $J_{\text{rung}}$ and $J_{\text{leg}}$ represent the coupling strengths between the legs and within the legs. Lastly, $\Delta$ denotes the XXZ anisotropy strength. Periodic boundary conditions are assumed. In the Hermitian limit, $\Phi = \Psi = 0$, the ground-state phase diagram has been studied in the literature. [50] We will mainly focus on the entanglement Hamiltonian in the rung-singlet phase. The schematic representation of the non-Hermitian spin-1/2 ladder is shown in Fig. 1.

Due to the non-Hermitian nature of the system, where $H^{\dagger} \neq H$, the time evolution of the wavefunctions is driven by both $H$ and $H^{\dagger}$ simultaneously. This results in the following time evolution equations: $\frac{\partial}{\partial t} |\varphi^{R}(t)\rangle = -iH |\varphi^{R}(t)\rangle$, and $\frac{\partial}{\partial t} |\varphi^{L}(t)\rangle = -iH^{\dagger} |\varphi^{L}(t)\rangle$. Note that $\hbar \equiv 1$ is set. Consequently, in analogy to standard linear algebra, the eigenvectors of a non-Hermitian Hamiltonian are generalized into biorthogonal left and right eigenvectors,

satisfying the following eigenvalue equations: $H^\dagger|\psi_n^L\rangle = E_n^*|\psi_n^L\rangle$ and $H|\psi_n^R\rangle = E_n|\psi_n^R\rangle$, with the biorthonormal condition: $\langle\psi_n^L|\psi_m^R\rangle = \delta_{nm}$. [24] In non-Hermitian quantum mechanics, observables are defined through the expectation value $\langle\psi^L|O|\psi^R\rangle$, involving both left and right eigenvectors. This biorthogonal framework naturally extends to the definition of the reduced density matrix in a bipartite system. For a system divided into subsystems A and B, the reduced density matrix (RDM) of the ground-state for subsystem A is defined as

$$\rho_A = \text{Tr}_B|\psi_0^R\rangle\langle\psi_0^L|. \tag{3}$$

Note that $\text{Tr}[\rho_A] = 1$, since $\langle\psi_0^L|\psi_0^R\rangle = 1$.

This biorthogonal definition of the RDM immediately raises a key issue: the RDM itself becomes non-Hermitian, meaning that its eigenvalues, $\omega_i$, are generally complex. Even when the total Hamiltonian has PT symmetry and the ground-state energy is real [22], the eigenvalues of the RDM in typical cases remain complex. As a result, appropriate definitions of generic entanglement entropy of both von Neumann type and Rényi type have been proposed by Tu, Tzeng and Chang to account for these complex eigenvalues. [28]

$$\begin{aligned}
S_{\text{TTC}} &= -\sum_i \omega_i \ln|\omega_i|, \\
S_{\text{TTC}}^{(n)} &= \frac{1}{1-n}\ln\left(\sum_i \omega_i|\omega_i|^{n-1}\right)
\end{aligned} \tag{4}$$

These entropies Eq.(4) effectively capture the negative central charge in non-Hermitian critical systems through the logarithmic scaling. [28, 29]

The entanglement Hamiltonian $H_E$ defined by $\rho_A = e^{-H_E}/Z$ is also non-Hermitian, and the entanglement energy $\xi_i$ is complex in general. The real part of the entanglement energy can be obtained directly as $\text{Re}[\xi_i] = -\ln|\omega_i| - \ln Z$, and the entanglement entropy Eq. (4) can be seen as the expectation value of the real part of the entanglement energy, $S_{\text{TTC}} = \sum_i \omega_i\text{Re}[\xi_i] + \ln Z$.[1] However, the imaginary part of the entanglement energy cannot be easily determined, as the logarithmic function becomes multi-valued. To gain further insight into this complexity, in the following section, we directly derive the entanglement Hamiltonian of the non-Hermitian spin-1/2 ladder using perturbation theory for the rung-singlet phase, where the ground state is adiabatically connected to the limit case of $J_{\text{rung}} \gg 1$.

## 3 Entanglement Hamiltonian

### 3.1 Perturbation Theory

The non-Hermitian spin-1/2 ladder Hamiltonian is given by Eq.(2). In the limit of $J_{\text{rung}} \gg 1$, the interaction between legs A and B at each rung $j$ defines the unperturbed Hamiltonian $H_0 = H_{AB} = \sum_{j=1}^{N} h_0^j$, where

$$h_0^j = J_{\text{rung}}\left[\frac{1}{2}\left(e^\Phi S_{j,A}^+ S_{j,B}^- + e^{-\Phi}S_{j,A}^- S_{j,B}^+\right) + \Delta S_{j,A}^z S_{j,B}^z\right], \tag{5}$$

---

[1] In PT-symmetric non-Hermitian systems, if the bipartition does not break the PT symmetry, the reduced density matrix remains PT-symmetric, meaning that its eigenvalues $\omega_i$ are either real or come in complex conjugate pairs. Consequently, the partition function $Z$ is real.

121 and the Hamiltonians of the legs $A$ and $B$, $H_1 = H_A + H_B$, are treated as perturbation. The left
122 and right eigenvectors of the single-rung Hamiltonian Eq.(5) are

$$|s_j^x\rangle = \frac{1}{\sqrt{2}}\left(e^{\sigma(x)\Phi}|\uparrow\rangle_{j,A}|\downarrow\rangle_{j,B} - |\downarrow\rangle_{j,A}|\uparrow\rangle_{j,B}\right), \tag{6a}$$

$$|t_j^{+x}\rangle = |t_j^+\rangle = |\uparrow\rangle_{j,A}|\uparrow\rangle_{j,B}, \tag{6b}$$

$$|t_j^{0x}\rangle = \frac{1}{\sqrt{2}}\left(e^{\sigma(x)\Phi}|\uparrow\rangle_{j,A}|\downarrow\rangle_{j,B} + |\downarrow\rangle_{j,A}|\uparrow\rangle_{j,B}\right), \tag{6c}$$

$$|t_j^{-x}\rangle = |t_j^-\rangle = |\downarrow\rangle_{j,A}|\downarrow\rangle_{j,B}, \tag{6d}$$

123 where $x = L, R$ labels the 'left' and 'right', respectively, and

$$\sigma(x) = \begin{cases} +1, & \text{if } x = R, \\ -1, & \text{if } x = L. \end{cases} \tag{7}$$

124 Let $|\psi_0^{L(0)}\rangle$ and $|\psi_0^{R(0)}\rangle$ denote the left and right ground states of the unperturbed Hamiltonian
125 $H_0$, with the corresponding ground state energy $E_0^{(0)} = -NJ_{\text{rung}}(\frac{1}{2} + \frac{\Delta}{4})$. Here, we assume
126 $\Delta > -1$. The ground state is a product of singlet states on each rung,

$$|\psi_0^{x(0)}\rangle = \bigotimes_j |s_j^x\rangle, \tag{8}$$

127 Using first-order perturbation theory, the corrected ground state can be written as,

$$|\psi_0^x\rangle \approx |\psi_0^{x(0)}\rangle + |\psi_0^{x(1)}\rangle, \tag{9}$$

128 where the left and right first-order correction terms are,

$$\langle\psi_0^{L(1)}| = \sum_{j=1}^N \sum_{n\neq 0} \frac{\langle\psi_0^{L(0)}|H_1|\psi_n^{jR(0)}\rangle}{E_0^{(0)} - E_n^{(0)}} \langle\psi_n^{jL(0)}|$$
$$= \frac{J_{\text{leg}}}{4J_{\text{rung}}} \sum_{j=1}^N \left[ \frac{2e^{-\Phi}e^{-\Psi}}{1+\Delta}...\langle t_j^+|\langle t_{j+1}^-|... + \frac{2e^{-\Phi}e^{\Psi}}{1+\Delta}...\langle t_j^-|\langle t_{j+1}^+|... - \Delta...\langle t_j^{0L}|\langle t_{j+1}^{0L}|... \right], \tag{10a}$$

$$|\psi_0^{R(1)}\rangle = \sum_{j=1}^N \sum_{n\neq 0} |\psi_n^{jR(0)}\rangle \frac{\langle\psi_n^{jL(0)}|H_1|\psi_0^{R(0)}\rangle}{E_0^{(0)} - E_n^{(0)}}$$
$$= \frac{J_{\text{leg}}}{4J_{\text{rung}}} \sum_{j=1}^N \left[ \frac{2e^{\Phi}e^{\Psi}}{1+\Delta}...|t_j^+\rangle|t_{j+1}^-\rangle... + \frac{2e^{\Phi}e^{-\Psi}}{1+\Delta}...|t_j^-\rangle|t_{j+1}^+\rangle... - \Delta...|t_j^{0R}\rangle|t_{j+1}^{0R}\rangle... \right]. \tag{10b}$$

129 Where, the dots represent the singlet states on each rung. $|\psi_n^{jL(0)}\rangle$ and $|\psi_n^{jR(0)}\rangle$ denote the
130 excited states of the unperturbed Hamiltonian $H_0 = H_{AB}$, and $E_n^{(0)}$ are the corresponding
131 energies. Specifically, the excited states with non-zero contributions to the corrections are

$$|\psi_1^{j(0)}\rangle = ...|t_j^+\rangle|t_{j+1}^-\rangle...,$$
$$|\psi_2^{j(0)}\rangle = ...|t_j^-\rangle|t_{j+1}^+\rangle...,$$
$$|\psi_3^{jx(0)}\rangle = ...|t_j^{0x}\rangle|t_{j+1}^{0x}\rangle..., \tag{11}$$

where $x = L, R$, and the corresponding eigenenergies are

$$E_1^{(0)} = E_2^{(0)} = J_{\text{rung}}\left[(1+\Delta) - N\left(\frac{1}{2} + \frac{1}{4}\Delta\right)\right],$$
$$E_3^{(0)} = J_{\text{rung}}\left[2 - N\left(\frac{1}{2} + \frac{1}{4}\Delta\right)\right]. \tag{12}$$

The reduced density matrix $\rho_A$ defined in Eq. (3) can be approximated as,

$$\rho_A \approx \rho_A^{(0)} + \rho_A^{(1)} = \text{Tr}_B\left(|\psi_0^{R(0)}\rangle\langle\psi_0^{L(0)}| + |\psi_0^{R(1)}\rangle\langle\psi_0^{L(0)}| + |\psi_0^{R(0)}\rangle\langle\psi_0^{L(1)}|\right)$$
$$= \frac{1}{2^N}\left(1 - \frac{4J_{\text{leg}}}{J_{\text{rung}}(1+\Delta)}\sum_{j=1}^{N}\left[\frac{1}{2}\left(e^{\Psi}S_{j,A}^{+}S_{j+1,A}^{-} + e^{-\Psi}S_{j,A}^{-}S_{j+1,A}^{+}\right) + \frac{1}{2}\left(\Delta + \Delta^2\right)S_{j,A}^{z}S_{j+1,A}^{z}\right]\right) \tag{13}$$

Thus, the reduced density matrix can be written in terms of the Hamiltonian of subsystem A,

$$\rho_A \approx \frac{1}{Z}\exp[-\beta\tilde{H}_A] = \frac{1}{Z}\left(1 - \beta\tilde{H}_A + \frac{1}{2!}\beta^2\tilde{H}_A^2 - \frac{1}{3!}\beta^3\tilde{H}_A^3 + \cdots\right) \tag{14}$$

Compare Eq. (14) with Eq. (13), we obtain the *ad hoc* inverse temperature

$$\beta = \frac{4}{1+\Delta}\frac{1}{J_{\text{rung}}} \ll 1, \tag{15}$$

and the Hamiltonian of the subsystem A

$$\tilde{H}_A = J_{\text{leg}}\sum_{j=1}^{N}\left[\frac{1}{2}\left(e^{\Psi}S_{j,A}^{+}S_{j+1,A}^{-} + e^{-\Psi}S_{j,A}^{-}S_{j+1,A}^{+}\right) + \tilde{\Delta}S_{j,A}^{z}S_{j+1,A}^{z}\right], \tag{16}$$

which is in the form of XXZ interaction with a renormalized parameter $\tilde{\Delta} = \frac{1}{2}(\Delta + \Delta^2)$. The partition is $Z = \text{Tr}[\exp(-\beta\tilde{H}_A)] = 2^N$.

## 3.2 Discussion

We make some remarks regarding the derivation: For the spin ladder Eq. (2) considered in this paper, the renormalized anisotropy parameter remains unchanged, i.e., $\tilde{\Delta} = \Delta$, when $\Delta = 1$ or $0$. In these specific cases, the entanglement Hamiltonian is exactly equal to the subsystem Hamiltonian, $\tilde{H}_A = H_A$. When $\Psi = \Phi = 0$, our results are consistent with those from earlier studies [16], reproducing the known Hermitian case. Even when non-Hermitian couplings are introduced with non-zero $\Psi$ and $\Phi$, the overall behavior remains remarkably similar, confirming that the methods and conclusions from the Hermitian regime can be successfully extended to non-Hermitian systems without major deviations. In the general non-Hermitian case, if one wishes to ensure that $\tilde{H}_A = H_A$, a simple approach is to choose the inter-subsystem Hamiltonian $H_{AB}$ as a Hermitian and isotropic Heisenberg interaction.

$$H_{AB} = J_{\text{rung}}\sum_{j=1}^{N}\vec{S}_{j,A}\cdot\vec{S}_{j,B} \tag{17}$$

This allows the reduced density matrix to reflect the exact form of the subsystem Hamiltonian without any parameter renormalization. A particularly interesting scenario arises when $H_A$ and $H_B$ are both Hermitian, while only the inter-subsystem coupling $H_{AB}$ is non-Hermitian, specifically with parameters $\Psi = 0$ and $\Phi \neq 0$. In this case, despite both the total Hamiltonian and the reduced density matrix being non-Hermitian, all the entanglement energies remain real. This situation is quite rare and demonstrates an unusual interplay between Hermitian and non-Hermitian components in the system.

## 4  Conclusion

The entanglement Hamiltonian, whose eigenvalue spectrum is known as the entanglement energy spectrum [5], plays a crucial role in revealing quantum correlations between subsystems in many-body systems. Understanding its analytical form is essential for gaining deeper insights into the nature of quantum entanglement and for facilitating entanglement Hamiltonian tomography [51]. However, obtaining the entanglement Hamiltonian is often challenging, especially in non-Hermitian systems, where it can be complex and non-Hermitian itself. While studies on non-interacting systems, such as the non-Hermitian Su-Schrieffer-Heeger (SSH) model [32,52], have made progress in deriving the entanglement Hamiltonian, the challenge is even more pronounced in interacting systems, where many-body effects add significant complexity. Although the real part of the entanglement energy can be derived from the eigenvalues of the reduced density matrix, capturing the full entanglement Hamiltonian remains crucial for exploring the deeper properties of quantum many-body systems.

In this paper, we explore the entanglement Hamiltonian of a non-Hermitian spin ladder system using perturbation theory, providing an analytical approach to this difficult problem. Remarkably, we find that the entanglement Hamiltonian in the non-Hermitian case can be approximated by the Hamiltonian of subsystem A, indicating that the thermal density matrix of an isolated non-Hermitian system A in equilibrium can be derived by the partial trace of an enlarged system. This suggests that the ancilla trick applied for developing finite-temperature Density Matrix Renormalization Group (DMRG) method [10], can be extended to non-Hermitian many-body systems as well. Our work offers new insights into the study of quantum entanglement in non-Hermitian systems, potentially facilitating the development of advanced numerical algorithms for investigating their finite-temperature behavior.

Although non-Hermitian systems exhibit many phenomena absent in Hermitian systems, such as exceptional points (EP) and the non-Hermitian skin effect, many features of Hermitian systems persist in non-Hermitian counterparts when appropriately generalized. For instance, the entanglement entropy in critical systems still follows a logarithmic scaling [28,29], fidelity susceptibility diverges near phase transitions or EPs [39,40], and machine learning methods can be transferred from Hermitian to non-Hermitian systems [53]. In our work on entanglement Hamiltonians, we extend the Hermitian case to non-Hermitian systems and find that, for nearly maximally entangled states, the results remain consistent with the Hermitian case.

## Acknowledgements

YCT is grateful to Chia-Yi Ju, Po-Yao Chang and Gunnar Möller for many invaluable discussions. We thank to National Center for High-performance Computing (NCHC) of National Applied Research Laboratories (NARLabs) in Taiwan for providing computational and storage resources.

**Funding information**   YCT is grateful to the supports from National Science and Technology Council (NSTC) of Taiwan under grant No. 113-2112-M-A49-015-MY3.

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
