# Peer review of "Entanglement Hamiltonian and effective temperature of non-Hermitian quantum spin ladders"

_SciPost Physics Core_

## Round 1 · Referee Report · Anonymous (Referee 1) · 2024-10-23

Report

The authors investigate the entanglement Hamiltonian of a non-Hermitian XXZ model on a ladder, focusing on the regime where the XXZ coupling between the two chains is much larger than the coupling within each chain. Using first-order perturbation theory, they demonstrate that the ground state entanglement Hamiltonian of a single chain is proportional to the chain's Hamiltonian, albeit with a modified XXZ anisotropy parameter.

While I don't find this result particularly surprising or groundbreaking, it is nonetheless a technical contribution that could merit publication in a specialized journal such as SciPost Physics Core.

I only have one comment that the author should address. Among the consequences of their result they say it "suggests that the ancilla trick applied for developing finite temperature Density Matrix Renormalization Group (DMRG) method, can be extended to non-Hermitian many-body systems as well". I am a bit confused by why they say so, since the "ancilla trick", as they call it, is a formal identity that enables the computation of thermal expectation values from the DMRG imaginary time evolution of pure states. It is widely used in finite-temperature calculations for 1-dimensional interacting systems and its applicability (in both Hermitian and non-Hermitian systems) does not rely on the fact that the ground state entanglement Hamiltonian of a ladder is proportional to the Hamiltonian. Therefore, I believe the authors should either elaborate more on this motivation or just cut it from the paper.

Recommendation

Ask for minor revision

  • validity: -
  • significance: -
  • originality: -
  • clarity: -
  • formatting: -
  • grammar: -

Author:  Yu-Chin Tzeng  on 2024-10-27  [id 4904]

(in reply to Report 1 on 2024-10-23)

We appreciate Referee 1's comments regarding the relevance of the ancilla trick in the context of our work. In response, we have removed the original discussion related to the finite-temperature DMRG method. Instead, we have reintroduced the ancilla trick as a well-established technique for studying finite-temperature properties, emphasizing its foundation in using maximally entangled states and imaginary time evolution. We now clarify that the proportionality of the entanglement Hamiltonian to the system Hamiltonian provides an alternative approach for exploring finite-temperature behavior. This alternative method could inspire experimentalists to design real systems that effectively simulate these finite-temperature properties. Our main work extends the knowledge from the Hermitian case to the non-Hermitian case, while the conclusion remains unchanged. We hope this addresses your concerns and strengthens the manuscript. Thank you once again for your valuable input.

---

## Round 1 · Referee Report · Anonymous (Referee 2) · 2024-10-25

Strengths

The main calculation is correct and well explained, reporting all details necessary to reproduce it.

Report

The authors study a non-Hermitian Hamiltonian, a ladder of two coupled XXZ chains with non reciprocal interaction.
They consider the limit of strong coupling between the two chains (the "rungs" of the ladder), treating the intrachain interaction as a perturbation.
At first order in perturbation theory, the authors show that, tracing out one of the two chains, the entanglement Hamiltonian of the other one in the ground state of the ladder is in form analogous to the non-reciprocal XXZ Hamiltonian of the single chain, with a "renormalised" anisotropy parameter $\widetilde{\Delta} = \frac{1}{2}\left ( \Delta + \Delta^2 \right)$ and the same non-reciprocity $e^{\Psi}$.

The result of this paper is analogous to what happens in the Hermitian case in the reciprocal XXZ ladder at strong "rung" coupling.

The main computation is simple and well explained, presenting all the details necessary to reproduce the computation.

I recommend the publication in SciPost Physics Core.

Recommendation

Publish (meets expectations and criteria for this Journal)

---

## Editorial Decision

resubmitted